# Navigating Cultural Challenges in Transplant Surgery: Insights from Turkish Surgeons

**DOI:** 10.3390/healthcare12131252

**Published:** 2024-06-24

**Authors:** Hicran Karataş, Şener Balas

**Affiliations:** 1Sociology Department, Faculty of Letters, Kutlubey Campus, Bartın University, 74100 Bartın, Türkiye; 2Ankara Etlik City Hospital, 74100 Ankara, Türkiye; senerbalas@yahoo.com

**Keywords:** organ transplantation, transplant surgery, transplant surgeons, cultural competency, cultural barriers

## Abstract

To achieve expertise, transplant surgeons in Turkiye undergo rigorous training, including medical school, residency, compulsory service, and extensive training in transplant surgery. Despite their high academic and clinical knowledge level, success in transplant surgery heavily depends on cultural competency. Through semi-structured interviews with 21 transplant surgeons specializing in kidney and liver transplants in Ankara, this study reveals how health illiteracy, culture, and folklore create significant barriers. Surgeons navigate these challenges while enduring harsh working conditions. This research highlights the critical role of cultural competency in transplant surgery, emphasizing the necessity for surgeons to understand and address the diverse cultural needs of their patients. Key findings indicate that surgeons must balance medical expertise with cultural sensitivity to deliver effective care. This study identifies four main cultural barriers: spiritual trust, family politics, health illiteracy, and subcultural incompetency. Effective transplant surgery requires a combination of theoretical proficiency and cultural awareness to meet a patient’s needs and improve surgical outcomes.

## 1. Introduction

Transplant surgeons complete medical school (6 years), residency (5 to 6 years), compulsory state service (500 days to 550 days), and transplant surgery training (24 months) to achieve the theoretical and clinical knowledge to practice transplant surgery. Their work environment is not limited to the hospital where they work, as they often must travel to very distant locations on short notice to secure their patients’ needs. Since organ procurement efforts require them to harvest deceased organs, which are very fragile and perishable, they must travel on short notice to perform their profession. Studies show that transplant surgeons are exposed to high levels of work-related risks due to the urgent nature of travel in organ procurement. According to Englesbe and Merion’s account, transplant surgeons focus on the urgency and importance of the task when they are called upon for organ procurement trips, and most of the time they take their safety for granted. Since time is of the essence in organ procurement trips, many organ transplant surgeons are involved in fatal accidents [1]. None mentioned the health risks posed by sleep deprivation experienced during organ procurement efforts. Our data showed that transplant surgeons often go without sleep for an average of 72 h from the moment of being called for an organ procurement trip until the moment the harvested organ is placed into the patient in need and the first follow-up is completed. Even though clinical studies define sleep deprivation as a significant health risk hazard to emotional state and mood, neurocognitive functions, executive functioning, decision-making, and physical health, the risk has not been identified among the work-related risks that transplant surgeons endure [2]. Nevertheless, transplant surgeons are expected to perform efficiently during harvesting and transplant operations despite the sleep deprivation produced by organ procurement trips.

Furthermore, the uniqueness of transplant surgery requires well-organized teamwork; highly skilled transplant surgeons are likely to travel with organ transplant perfusionists, attending surgeons, surgical nurses, and trained personnel. Their work forces them to be ready 24/7, which makes it challenging to organize their personal life. Many abandon their practice, for which they spent years to reach clinical excellence, because their job forces them to demand too much from their families [3,4,5,6,7]. Indeed, transplant surgeons reported working 65 to 70 h per week, with a median of 195 operative cases per year [3,4,8,9]. Despite the complexity of its performance, transplant surgery only offers surgeons noncompetitive salaries. Financial dissatisfaction is reported as one of the primary reasons for abandoning transplant surgery. Studies showed that transplant surgeons conduct many unbillable activities. Their time and effort are not reflected in their salaries as they often perform transplant surgery after hours.

Their salaries include on-call obligations and urgent organ procurement trips [3,4,7,10]. Several studies also reported that transplant surgery does not offer career planning for its practitioners. While many have difficulty collecting data, others complain that they spend too much time in practice and cannot harvest available data for publication [3,4,6]. This study explored the impact of cultural competency on transplant surgeons’ success in Turkiye. By understanding how cultural barriers affect surgeons and their ability to provide effective patient care, this study sought to offer insights that could improve patient outcomes. Addressing these challenges through enhanced cultural competency can help surgeons navigate the complex interplay between medical expertise and cultural sensitivity, ultimately improving patient care.

## 2. Method

### 2.1. Participants

We designed an ethnographic fieldwork study to understand the cultural dimensions of organ transplantation and the issue of insufficient organ donations. We engaged extensively with the social actors whose lives revolve around organ transplantation and donation, including patients, caregivers of patients, donor families, organ procurement coordinators, and transplant surgeons. During our fieldwork from 15 August 2022 to 15 October 2023, we collaborated with 21 transplant surgeons specializing in kidney and liver transplantation. The National Organ Transplant Center in Ankara operates nine Regional Organ Coordination Centers (RCCs) throughout Turkiye, including the Ankara RCC. There are ten organ transplant centers in Ankara where kidney and liver transplants are performed.

Most of our participants (*n* = 18) were affiliated with state-owned organ transplant centers, while the remainder were associated with organ transplant centers linked to private university hospitals. Among our participants, a significant portion (*n* = 16) held academic positions, primarily as professors and associates, with a few others (*n* = 3) progressing in their academic careers. The ages of our participants varied, with the youngest being 37 years old and the eldest being 70 years old. Most respondents were male (*n* = 18), with a smaller proportion being female (*n* = 3). Regarding marital status, most participants (*n* = 19) were married, while two were divorced. All informants were parents (Appendix A).

### 2.2. Data Collection

We facilitated semi-structured, in-depth, face-to-face interviews to explore whether surgeons encountered particularly challenging situations with their patients that were solely influenced by culture and folklore and how they handled these challenges to meet patient needs. Semi-structured interviews covered several themes: (1) how surgeons decide to pursue a career in transplant surgery, (2) how their profession affects their personal life, (3) how patients’ health literacy, particularly in the area of organ transplantation, challenges their practice, (4) whether transplant surgeons encounter problems solely caused by social norms, folk beliefs, and the religious views of patients, and (5) how they address these challenges. The informants’ responses provided personal experience narratives, highlighting how cultural competency significantly impacts the success of transplant surgery. Additionally, we recorded observation notes at the end of each day spent in organ transplant centers. Our engagement with surgeons involved closely following their daily schedules over the course of a year, commencing at 6:30 in the morning and concluding when they departed from the hospital premises. We accompanied them through clinical activities such as patient consultations, medical assessments, and interdisciplinary meetings, and conducted in-depth interviews with healthcare providers and recipients.

### 2.3. Data Analysis

All recorded interviews were transcribed using Transcriptor and reviewed by the research team to ensure accuracy. During the translation of the transcribed records, we carefully preserved the cultural and emotional nuances expressed by our informants. The responses of informants and observation notes regarding interview questions matched. After reading transcripts from several interviews, the authors developed a coding scheme to capture the essence of the testimonials using MAXQDA 2024. We facilitated thematic analysis to identify and report the patterns that emerged in our data. The theme of cultural barriers emerged with four sub-themes: spiritual trust, familial politics, health illiteracy, and subcultural competency.

### 2.4. Ethical Considerations

Due to the highly personal nature of the interview contexts, strict confidentiality measures were implemented for our informants. The actual identities of the surgeons, as documented on consent forms and audio recordings, are known solely to the research team. To safeguard anonymity, all real names of informants were replaced with common Turkish names, ensuring confidentiality between the research team and participants. By providing anonymity, our informants felt free to express their thoughts, feelings, and personal experiences, which may have influenced their performance.

Furthermore, in line with our ethical obligations, we took measures to safeguard their social and economic well-being during and after the fieldwork. Studies have demonstrated that ensuring anonymity for informants in qualitative studies not only fosters trust between researchers and participants but also encourages informants to share their experiences without fear of repercussions related to the research topic [11,12]. Since most participating surgeons voluntarily engaged in our research within state-owned hospitals and held academic positions, our study posed no threat to their well-being. Before embarking on fieldwork, ethics committee approval was obtained from the Bartın University Social and Humanities Research Ethics Board, 2022 SBB-0055, on 22 February 2022.

## 3. Results

The findings of this study highlight several critical cultural barriers. The first central theme was spiritual trust, where many patients express profound trust in their surgeons, often invoking religious beliefs by stating, “I entrust my life first to Allah and then to you.” This expression adds emotional weight to the surgeons’ responsibilities and increases their stress during transplant operations. Most respondents (*n* = 18) navigated this emotional burden by placing their faith in Allah for the outcome while performing to the best of their abilities, with only a minority (*n* = 2) engaging in discussions about potential outcomes with patients during the consent process to manage this stress. Family politics also play a crucial role, with most respondents (*n* = 20) expressing frustration over fully informed and consenting prospective donors expecting surgeons to liberate them from donation without disclosing the truth to the prospective recipient. This dynamic is particularly challenging when female family members are expected to donate, leading surgeons (*n* = 13) to take extra care to ensure donors are fully informed and not coerced.

Health illiteracy among patients and the general public presents significant challenges, as all interviewees (*n* = 21) noted patients’ tendencies to trust unreliable sources over medically accurate knowledge. This leads to unrealistic expectations and misunderstandings. Health illiteracy also complicates organ procurement trips, with several informants (*n* = 11) highlighting issues that take place during commercial airplane journeys. Lastly, subcultural incompetency was evident as surgeons encounter various subcultural norms impacting their practice. Jehovah’s Witnesses refuse blood transfusions due to religious beliefs, compelling surgeons (*n* = 6) to devise strategies like pre-storing patients’ blood.

In contrast, Alevi-Bektashi patients face challenges related to ritual kinship, which is often unrecognized by legal regulations. The majority of informants (*n* = 11) indicated that Alevi patients felt offended when doctors questioned their ritual brotherhood. These subcultural nuances necessitate surgeons to deeply understand their patients’ cultural backgrounds to provide adequate care. This study underscores that cultural competency is crucial for transplant surgery success, and is often gained through practical experience rather than formal training, highlighting the need for ongoing cultural education in medical practice.

### 3.1. Working with Patients Who Entrust Their Lives First to Allah and then to the Surgeon

Physicianship, as a profession, necessitates practitioners to determine the best course of action for patients whose lives are disrupted by illness. Their involvement in human life imbues the profession with a sense of sanctity. This sacred status can be traced back to the healing practices of shamans, who served as society’s physicians during pre-Islamic times among the Turks [13]. With the advent of Islam, the sacredness of healing and healers was further reinforced. Islam emphasizes the sanctity of physicians, as evidenced by the 32nd Ayat of Surah Al Maida, which states, “...if anyone saved a life, it would be as if he saved the whole people...”

Medicine is considered a communal responsibility, known as Fard Kifayah, aiming to “attain benefit or prevent harm.” Physicians fulfill this collective obligation by providing healing, thereby relieving other members of society from the religious responsibility of healing. In Islamic teaching, physicians are regarded as earthly practitioners and are seen as embodiments of one of the names of Allah, Al-Shafi (The Healer). [14].

Patients often convey their profound trust in physicians with the statement, “I entrust my life first to Allah, then to you,” thereby endowing physicians with a sacred responsibility in their practice. Alongside the ethical and professional obligations towards their patients, physicians grapple with managing these religious responsibilities articulated by patients. All respondents indicated that this expression adds to the stress they already experience during transplants. Consequently, surgeons have developed strategies to cope with the emotional weight imposed by patients who express transcendent trust before undergoing a transplant. Most respondents (*n* = 18) mentioned that they navigate this emotional burden by placing faith in Allah for the outcome of the operation while giving their utmost effort. To illustrate, Dr. Ahmet said that "Who will live, or die is up to Allah. I only operate to the best of my ability. However, the decision of whether my patient will be cured or not is determined by Allah. I perform well, but outcomes cannot be guaranteed before or during the operation. Sometimes, everything goes well, but the patient rejects the kidney. It happens. As long as I do my best, that is what matters.”

Only a minority (*n* = 2) stated that engaging in discussions about potential outcomes with patients during the consent process and prioritizing excellence in the operating room serve as coping mechanisms to alleviate the anxiety induced by the profound trust of the patient. Dr. Emir exemplified this approach, stating, “I present all possible risks that can happen during my operation to both the recipient and donor. I want to ensure they are fully informed about the transplantation procedure and potential outcomes. They often express, ‘There is Allah above; below you’ (Tr. Yukarıda Allah, aşağıda sen), which underscores the immense responsibility felt upon entering the operating room. However, informing them about the risks of the operation helps me navigate the delicate balance between emotions and practice.” Observing Dr. Emir’s practice, it became evident that his invocation of “bismillah” (in the name of Allah) before commencing an operation was indicative of the stress triggered by the profound trust placed upon him by his patients. Only one respondent emphasized that having a highly experienced supervising surgeon in the operating room reduces emotional tension.

Medical students are not directly trained in cultural competency within the formal curriculum. can only be attained via face-to-face interaction between medical students and patients. They also become familiar with possible cultural matters that can affect their practice by listening to the personal stories of their supervisors, who have much experience in the field, which prepares them to be aware of the relationship between success and cultural competency. As Dr. Şeref stated, students listen to the stories regarding cultural challenges experienced by their supervisors, and they only have a limited idea of how crucial cultural competency is for securing medical practice’s success. During five years of training in surgery after six years in medical school and an additional two years of transplant training, cultural competency can be obtained only by experiencing particular cases. As Dr. Fatih stated, the duration of medical training for transplant surgeons does not ensure their excellence in practice, because culture is complex and central to the treatment process.. “I have been doing transplants for 12 years now. I am always surprised by the culture of the patient, which directly affects my operation and follow-ups. Mediating between the cultural needs of the patient and the required treatment always produces good results. While young and inexperienced, I confronted the cultural needs of patients, defending universal principles, which only made my practice more stressful.”

By entrusting life first to Allah and then to physicians, patients place physicians in a mediating role between Allah and themselves, expecting them to serve as earthly conduits of healing. Additionally, most respondents (*n* = 13) noted that when relatives of recipients or donors express trust through the expression above, it heightens the emotional responsibility physicians feel during operations.

Parents of pediatric patients undergoing transplants often express the most profound trust. Given that all of our respondents were parents, the experience of parenthood among surgeons and the weight of being entrusted with such responsibility were distressing factors during operations. Transplant surgeons, before graduation, are expected to demonstrate proficiency by serving as the principal surgeon in a minimum of 30 kidney and 45 liver transplants, in addition to receiving training in pre/post-operative management of transplant patients. Nevertheless, even the most experienced transplant surgeons operate under the heavy burden of trust. Dr. Aydoğan, with 42 years of experience in transplant surgery and having conducted thousands of kidney and liver transplants, reflects, “When a patient’s wife or mother entrusts their loved ones to me, it induces anxiety during the operation. If, despite my best efforts, I were to lose a patient, it would fill me with deep sadness. The patient’s trust in us makes us feel indebted, and we aim to repay this debt by healing the patient. However, sometimes, despite all our efforts, this cannot be achieved.All we can do in those moments is entrust our performance to Allah. There is simply no other way to navigate this emotion.”

### 3.2. Surgical Performance within the Realm of Familial Politics

Illnesses are not merely individual experiences; they also have social dimensions. While the patient struggles with the illness, the social environment is reshaped to meet the patient’s needs.Moreover, family members, friends, and others witness patients’ suffering. This is often manifested through societal expectations of sick roles. However, the interpretation of these sick roles is primarily influenced by cultural contexts. Society delineates the components of sick roles in terms of rights and responsibilities attributed to both the patient and their social circle. As Kleinman and Benson argued, culture serves as a framework through which even mundane activities and circumstances acquire emotional significance and moral implications for those involved [15]. In the context of transplantation, which offers hope to patients with organ failure by potentially freeing them from dysfunctional body parts, patients rely on their close relatives, often expecting organ donation.

Even though many studies have discussed how family members feel a moral obligation to donate their spare organs to the patients in a family circle without being aware of the risks posed by organ transplant operations, none have pointed out that surgeons might need to be mediators in family politics [16,17,18]. Most respondents (*n* = 20) expressed frustration that prospective donors, who are fully informed and consenting, expect surgeons to liberate them from donation without disclosing the truth to the prospective recipient. These donors do not initially wish to donate, yet they also fear potential resentment from the recipient if they refuse.

Some potential donors within the family circle visit organ transplantation centers (OTCs) to demonstrate the strength of familial bonds, hoping that test results will be negative. However, there are numerous instances where donors with positive results opt out of donation for various reasons, entrusting surgeons to relieve them of their familial obligation. In our research, we interviewed patients awaiting cadaveric organ donations, and the majority (20 out of 30) expressed resentment toward their family members for not offering their spare organs despite witnessing their suffering [19]. While safeguarding familial relationships between patients and potential donors may not be an ethical responsibility of surgeons, cultural norms compel them to act as mediators in family dynamics to shield the patient from further distress.

According to organ donation laws, blood relatives or relatives by marriage up to the fourth degree are eligible to donate spare organs. Cross-transplantation is also allowed with informed and consenting parties. However, non-related donor candidates undergo scrutiny by an ethics committee to ensure their altruistic motivations [20]. Consequently, patients often rely on close relatives for kidney or liver donations. However, even close family members may hesitate to undergo such invasive procedures. Glannon and Ross argued that while familial bonds create moral obligations for organ donation, this obligation can be overwhelmingly burdensome for family members [17]. Studies indicate that the expectation for family members to donate is rooted in the belief that intimate relationships generate moral obligations within families, a principle not unique to organ donation [16,17,18,19], which highlights the universal struggle within family relationships to balance selfishness and altruism, placing prospective donors in challenging positions.

In our fieldwork, consisting of interviews with patients awaiting organ donation (*n* = 30), living organ recipients (*n* = 30), and cadaveric organ recipients (*n* = 30), we observed that single or childless family members fall outside the scope of these expectations. Conversely, patients frequently rely on their parents and siblings who are married with children. Additionally, we noted that the working conditions of prospective donors also influence the patient’s expectation level. Housewives, in particular, are expected to step forward as donors since they are not employed, and other family members can assist them during their recovery period. When potential donors were female, transplant surgeons exercised extreme caution to ensure that they were not being coerced into donation. Most of our informants (*n* = 13) indicated that when patients brought their sisters-in-law, daughters-in-law, sisters, wives, or mothers as potential donors, they ensured that the donor was fully informed and willing to undergo the donation process. Dr. Oskay said:

“When the potential donor is female, we must proceed cautiously. I pay close attention to her demeanor during family meetings; her jests, facial expressions, and subtle cues often reveal her reluctance. In such cases, I provide both the recipient and donor with my business card, encouraging the donor to reach out and share her thoughts with me. More often than not, they contact me to express concerns about feeling pressured or fearful of offending the patient. In response, we work together to navigate a solution that respects her wishes and exempts her from the donation process”

It is well-known that in cost-free living donations, two-thirds of all organs are donated by women. Studies have related this to women’s sense of responsibility and tendency to self-sacrifice [21,22]. Dr. Ulvi’s experiences with female donors suggested that female members of the family who are parents and housewives are the first to be expected to be organ donors: “Male patients often behave as if their wives and elder sisters must sacrifice their kidneys or livers for the male members of the family. In many cases, I have witnessed older male patients bringing their daughters-in-law instead of their sons. They justify this by suggesting that their sons must work, while their daughters-in-law stay home, as if they do not have to take care of children or run errands for their families. I meticulously investigate to ensure that male patients are not coercing female donors.” Dr. Saniye, a female organ transplant surgeon, underlined that female donors often reveal their reluctance by showing disinterest in the information given at family meetings or, in some cases, by using gestures and facial expressions. “If female donors are reluctant, they do not ask questions about the procedure, they seem not caring about what will happen next, or they do not establish any eye contact. Most of the time, on the other hand, they wink at me in a way that makes me understand that there is something that is not right. When these happen, I find an excuse to be alone with the donor. They often feared the family would cast them out if they refused to be donors. I always find a way to get her out without letting the family know that she does not want to be a donor.”

While there is consensus in the literature that transplant surgeons must prioritize safeguarding the well-being of healthy individuals who choose to donate their spare organs out of altruism or a sense of moral duty, some scholars have noted that familial moral obligations can sometimes coerce reluctant individuals into donating as well [18,19,21,22,23]. Bioethical considerations surrounding organ transplantation often overlook the procedure’s social dimensions, focusing primarily on the health outcomes for patients and prospective donors. Emphasizing the moral responsibility of surgeons to decline both related and unrelated donors if the medical benefits do not justify risking the health of the donor, some scholars have also pointed out that donors, whether related or unrelated, may donate driven by a non-traditional sense of altruism [4,17,18,23]. Furthermore, non-traditional altruismin organ transplantation extends beyond familial moral obligations.In some cases, donors are pressured into donating. Families may marginalize potential donors who show reluctance [19]. Despite transplantation being grounded primarily in codified medical knowledge acquired through formal training, surgeons are also active members of society, allowing them to empathize with patients and donors. While they operate within a medical community where disease and treatment are often strictly defined by biomedical and bioethical principles, their personal experiences within families and patient interactions aid in overcoming cultural alienation.

### 3.3. Dealing with Health Illiteracy: “Did you Transplant the Donor’s Brain as well?”

Nearly ten months after a cadaveric donation, the family of the deceased initiated legal action against an individual who they believed had engaged in a physical altercation with the dead. According to the family’s account, the defendant allegedly struck the deceased’s head three weeks before the latter fell from a height, resulting in brain death. Initially classified as a workplace accident, compensation was provided by employment insurance promptly after the incident occurred, and negligence claims resulted in additional compensation for the family. However, dissatisfied with this outcome, the family sought further legal action upon learning that their relative had been involved in a confrontation with a coworker who allegedly kicked the deceased in the head. They asserted that this individual was responsible for the dead’s demise. Despite the initial autopsy report indicating no skull damage, the family persisted in requesting a second autopsy, which the court granted. Subsequently, the public prosecutor summoned transplant coordinators to ask for all pertinent documents related to the kidney and liver transplant procedures. After receiving the submitted documents, which included notes and operation records, the prosecutor interrogated the chief surgeons regarding whether the transplant team had conducted a brain transplant on the donor. Dr. Mehmet explained that this is an ordinary experience for a transplant surgeon since the layman’s understanding of transplant surgery is informed mainly by fictional movies, TV shows, and folklore.

All interviewees expressed frustration that their patients tended to trust information from unreliable sources rather than the medically accurate knowledge derived from clinical experience. Dr. Ali elaborated on this phenomenon, noting that transplant surgery is particularly complex, even for medical professionals outside the surgical field. He empathized with his patients, acknowledging their curiosity and desire to understand the intricacies of the transplant process. According to Dr. Ali, recipients often seek detailed information, sometimes through reading or watching videos about transplants. Conversely, donors, who are typically healthy individuals offering their organs to loved ones, are given priority. During family meetings, transplant surgeons provide comprehensive medical information about the procedure, encouraging questions and discussions about the operation and post-operative care. Dr. Ali emphasized that many donors only learn the specifics of the process during these meetings. Dr. Burhanettin observed that recipients often overlook the surgical procedure’s complexities, assuming donors quickly recover without significant risks. According to informants (*n* = 8), this perspective arises from the pain and anxiety experienced by patients during the search for a living donor. Dr. Ayşe elaborated on this, explaining that patients fear potential donors may change their decision upon learning about the risks associated with the healing process. This fear is often realized, as patients, overwhelmed by the wait for a cadaveric donation, may downplay the risks donors are willing to undertake to assist the recipients.

Transplant surgeons do not interact with the relatives in cases of brain death, who mainly communicate with intensive care doctors and organ transplant coordinators. The cultural complexity of brain death interferes with transplant surgeons’ practice, causing insufficient numbers of cadaveric organ donations. Reports indicated that 5268 transplants were conducted in 2022 in Turkiye, with 4802 of these donations originating from living donors [24]. Since the cultural understanding of death is mainly related to the heart, brain death as a medical term is relatively new to the layman. The fact that only 24 out of 289 cadaveric donations included hearts in 2022, despite 1302 patients awaiting heart donations, underscores the persistence of the traditional concept of death within the realm of cadaveric organ donation in Turkiye [24]. While the informants possessed expertise in cadaveric liver and kidney transplants, the shortage of cadaveric donations compels surgeons to primarily perform living organ transplantation.

Health illiteracy can also impact the performance of transplant surgeons during organ procurement. Regional organ coordination centers typically organize organ procurement trips. The National Organ and Transplantation Center also coordinates procurement trips between regional centers. Surgeons may travel by various means, depending on the distance between the donor hospital and the transplant center, and the case’s urgency, including in military planes, scheduled flights, helicopters, cars with drivers, personal vehicles, or ambulances.

Surgeons (*n* = 11) have reported that commercial airplane journeys pose significant challenges during organ procurement trips, often requiring them to address issues related to health illiteracy. The majority of informants complained that despite labeling the boxes carrying harvested organs with “human organ for transplantation,” they are not permitted in the cabins of commercial flights. Furthermore, security personnel often require surgeons to open the organ transportation boxes for inspection, questioning their contents.

Dr. Serdar summarized his experience: “After harvesting, we are supposed to be at the host hospital to perform transplantation within 24 h. Commercial trips are the most challenging since aircrew and security personnel are unaware that harvested organs are fragile. I often had to contact my superiors, who once called the company’s CEO, so security would allow me to pass through security doors without opening the box.” Several informants (*n* = 3) noted that passengers on the plane were disturbed by surgeons traveling with transplantation boxes. Most respondents stated that passengers do not prioritize allowing surgeons to exit the aircraft promptly, even though an ambulance was waiting to transfer the organs. The conditions and duration of organ procurement trips can significantly affect the durability of organs due to various stressors, including shifts in temperature and vibration. Studies have shown a correlation between the duration and conditions of procurement trips and the functional outcomes of transplantations, suggesting that smoother organ procurements promise more prolonged survival of transplanted organs [1,25].

### 3.4. Exploring Subcultures through Clinical Experience

Cultural beliefs, behaviors, and values significantly influence the diagnosis and effectiveness of treatments in medical practice. The experience of illness, healing, or enduring suffering often carries moral significance shaped by cultural norms. These factors infuse concepts of illness and treatment with contextual meanings shaped by variables such as age, gender, ethnicity, education, and social support networks (such as kinship and friendships). While Manassis revealed that patients’ help-seeking behavior is influenced by their perceptions of symptom severity, Zola’s findings suggested that symptom appearance or worsening may not always trigger medical attention-seeking behavior [26].

Patients may distrust young or female doctors in some cultural contexts, preferring traditional healers, often older members of their local community [27,28]. Additionally, as highlighted by Post and Weddington, culturally approved coping strategies can impact medical help-seeking behavior and physician treatment responses [29]. Cultural norms also establish expectations regarding the boundaries of health and create hierarchies of illnesses based on their social impact. Patients may dismiss specific symptoms as natural, varying among subcultures, while in other cases, culture may validate particular symptoms as indicators of serious illness, thereby affirming the patient’s sick role. As Parsons noted, communities are vested in minimizing illness, as illness can disrupt social constructions and impede individuals from fulfilling expected social roles [30]

Patients, socialized within their families, navigate cultural decisions surrounding symptom labeling, validation of the sick role, and responses to medical treatment, often through negotiations among physicians, patients, and family members [30,31,32,33,34,35]. While clinicians may overlook cultural dimensions impacting treatment outcomes as they approach disease and treatment through medical frameworks, understanding and appreciating cultural factors in diagnosis and treatment planning can enhance patient care [36]. However, it does not guarantee treatment outcomes. With its diverse subcultures coexisting for centuries, Turkiye provides a rich context for exploring these cultural dynamics in medical practice. Surgeons in this realm acknowledge that a patient’s cultural background profoundly influences all aspects of the operation, with some subcultures presenting unique challenges. Jehovah’s Witnesses and Alevi-Bektashi communities stand out as particularly complex cases.

For Jehovah’s Witnesses, blood holds sacred significance, and their religious beliefs prohibit them from accepting blood transfusions. Surgeons in surgical fields often seek non-operative solutions when feasible, resorting to synthetic blood if surgical intervention is unavoidable [37,38,39]. Our informants highlighted the extreme anxiety Jehovah’s Witness patients experience when faced with the prospect of transplantation, as they adamantly refuse blood transfusions during and after the operation. While traditional doctrine prohibits the harvesting of organs from humans, modern interpretations of religious teachings have empowered Jehovah’s Witness patients with organ failure to make decisions regarding transplantation, provided that blood transfusions are strictly avoided. However, this poses a significant clinical challenge, as these patients refuse potentially life-saving blood transfusions that may be necessary during the operation.

Given that transplantation procedures often involve significant pre-operative blood loss, Jehovah’s Witness patients present formidable challenges to surgeons. Regulations mandate that transplant teams obtain fully informed consent forms, including consent for potential blood transfusions and awareness of associated risks. Surgeons in state hospitals are often reluctant to operate on Jehovah’s Witnesses due to the high operational risks involved. In contrast, those working in foundation hospitals may encourage patients to store their blood in the blood bank to facilitate surgery.

Six of the interviewed surgeons recounted encounters with Jehovah’s Witness patients suffering from kidney or liver failure, recommending that these patients store their blood before scheduled transplants. To formalize this arrangement, patients and surgeons sign a contract stipulating that the surgeon will exclusively use the stored blood during the procedure, refraining from additional transfusions. Dr. Toygar described this practice: “When Jehovah’s Witnesses inquire about transplant options, I advise them to store a unit of blood in the bank. This allows me to utilize their blood when necessary. They request us to sign a contract guaranteeing no blood transfusions during the operation except their own. Our standard forms do not accommodate such arrangements, so we sign a separate contract to reassure patients that their cultural needs will be respected, even in life-threatening situations.”

In contrast, Dr. Burhan shared his approach, having performed transplants on three Jehovah’s Witness patients. He cautions patients that if the stored blood proves insufficient, he will not take the risk of jeopardizing their lives: “I cannot bear the thought of my patients losing their lives due to inadequate blood transfusions. That is why I only proceed with surgery for Jehovah’s Witnesses under the condition that their stored blood will be used. However, if this proves insufficient, I am prepared to assume responsibility for additional transfusions.”

Jehovah’s Witnesses present significant clinical challenges in transplant surgery and all surgical specialties. Research has highlighted that addressing the subcultural needs of patients requires physicians to devise culturally acceptable solutions to ensure effective treatment [38,39]. According to the surgeons interviewed, the most culturally appropriate solution is to have patients store their blood before surgery, enabling them to proceed with an organ transplant while adhering to their religious beliefs. Additionally, the informants noted that Jehovah’s Witnesses typically do not favor transplants from cadaveric donors, as they prefer genetically compatible organs, aligning with their religious convictions.

Another challenging cultural subgroup is the Alevi-Bektashi, followers of a sect within Islam. While Alevi patients are generally supportive of cadaveric organ donations, social norms within the community impose strict limitations on living donations. Interviews conducted with 21 dede, the religious leaders of Alevi communities, alongside insights gathered from transplant surgeons’ personal experiences, suggest that physicians with backgrounds in the Sunni tradition may face cultural challenges when treating Alevi patients.

The traditional Alevi understanding of death suggests that only the body perishes. At the same time, the deceased’s soul remains in the world until the Day of Judgment, striving to attain the highest spiritual state. A person’s death marks the beginning of their soul’s eternal journey toward the Truth (Tr. Hakk). Before embarking on this journey, an Alevi must purify their soul, relinquish worldly attachments, and obtain consent from all with whom their body has interacted. The Alevi faith is founded on mastering speech, actions, and thoughts. Since social discord can impede the soul’s posthumous journey through the individual’s speech, actions, and thoughts, Alevis must find contentment within themselves, and their social environment must also be content with them [40,41,42]. The Alevi faith promotes cadaveric organ donations since the spiritual station of the soul is elevated through the good deeds of prospective donors and the deceased’s donation.

On the other hand, Alevi social norms (Tr. Erkan) regulate living organ donation. A married, mature Alevi individual must establish an artificial brotherhood and embark on their journey into Alevism through the Jem ritual (Tr. Cem). A male, married aspirant (Tr. Talip) binds his life and belief through brotherhood. The Jem and another Alevi aspirant promise each other to maintain their brotherhood for a lifetime. This ritual is binding for family members of both parties as well. The ritual contract requires the aspirant to see his brother (musahip) as his equal in health and wealth. According to the ritual agreement’s content, members must prioritize each other’s well-being. Ritual brotherhood is more binding than blood brotherhood since the ritual agreement requires musahips to be equal in health and wealth. According to the Alevi faith, musahips must provide spare organs to each other when one is in need to restore equality in health states. The law regulating living organ donation, on the other hand, mandates that parties go through an ethical committee, which is heartbreaking for prospective recipients and donors.

The majority of informants (*n* = 11) indicated that Alevi patients felt offended when doctors questioned their ritual brotherhood. Dr. Bahriye shared her perspective, stating, “Alevi patients are susceptible to organ donation. I have not encountered any Alevi patients struggling to find donors. However, since recipients and donors lack official kinship ties, we must refer them to the ethical board. This decision deeply saddens them. It is due to legal requirements, but they interpret it as questioning their sincerity.” On the other hand, Dr. Serhat faced accusations of racism from his Alevi patients. They expressed anger and questioned his ethical integrity when he directed them to the ethical committee to verify the altruistic motives of both parties.

Ritual brotherhood, rooted in love and performed before a dede (spiritual leader of the Alevi faith) and witnesses, is a significant aspect of Alevi culture. Aspirants must know each other and be near to provide support, making their ritual brotherhood known to their community. However, despite this, an ethical committee review is necessary to confirm that there are no economic or personal interests between the parties beyond altruism. Although the ethical committee ultimately approves transplantation permits, the process can be overwhelming for Alevi patients and donors. Dr. Ali articulated the situation, saying, “Alevi patients and donors perceive our compliance with the law as a rejection of their faith. We are simply following legal requirements to ensure the donor’s safety. I cannot risk my career as a surgeon by disregarding the need for an ethical board permit.” Dr. Yusuf noted that some Alevi patients requested to be directed to transplant centers with Alevi surgeons to avoid ethical investigations. However, regardless of religious adherence, surgeons are bound by regulations and cannot proceed without the necessary permits.

## 4. Discussion

Given the cultural diversity in Turkiye, physicians regularly interact with patients from various ethnic backgrounds and cultural heritages. The effectiveness of medical care is not solely dependent on the theoretical proficiency of practitioners but also on their ability to address the cultural needs of patients. Culture shapes patients’ expectations and influences their attitudes and behaviors towards treatment. Western medical training often assumes that all patients will be highly educated and compliant with prescribed medicines. However, doctors encounter many diverse culture-nurtured scenarios in their daily practice. Transplant surgery is a distinctive medical specialty wherein extensively trained surgeons perform the pivotal task of replacing malfunctioning organs with healthy counterparts. This transformative intervention bestows renewed vitality upon patients and is a beacon of hope, easing their burdensome afflictions. While extensive research has delved into the professional challenges faced by transplant surgeons, only a limited number of studies have explored whether these professionals encounter difficulties arising from the cultural needs of their patients [4,6]. To address this gap, we augmented our semi-structured, face-to-face interviews with transplant surgeons by incorporating additional inquiries to investigate the potential correlation between the cultural competency of surgeons and the success of transplant surgery. Our results identified four themes: spiritual trust, family politics, health illiteracy, and subcultural incompetence.

Central to our study was the profound trust that patients place in transplant surgeons, often invoking religious beliefs and entrusting their lives to both Allah and the physician. This expression of trust not only amplifies the emotional responsibility felt by surgeons but also shapes their coping mechanisms and approach to patient care. Dr. Ahmet’s perspective exemplified the delicate balance between professional competency and reliance on divine intervention, highlighting the existential uncertainties inherent in surgical practice.

While many have highlighted the moral obligation placed on family members in the context of organ transplantation to donate spare organs, there has been a notable absence of mentions regarding the involvement of transplant surgeons in family politics to alleviate this emotional burden in morally obligated prospective donors [16,17,18]. Turkish transplant surgeons are dragged into family politics by reluctant living organ donors who are afraid of confrontation or being marginalized by family members by labeling them with selfishness. As Elliot pointed out, a physician’s role entails assessing the reasonableness or moral justifiability of a patient’s decision and determining if the physician is morally justified in aiding the patient in achieving it [43]. In such instances, the participation of surgeons in family politics is geared toward safeguarding hesitant living organ donors. Moreover, as moral agents, surgeons reserve the right to decline a donor if the associated risks are deemed excessively high, particularly in emotionally charged donation scenarios [17,18]. Transplant surgery’s complexity is often misunderstood by those outside the medical field, with the layman’s knowledge largely shaped by urban narratives and fictional movies. Dr. Ali’s accounts illustrated how medical professionals outside transplant surgery may have difficulty understanding the details of transplant surgery. The impacts of health illiteracy were mostly traced through the personal experience of transplant surgeons on organ procurement trips. Even though studies have underlined the risks of these trips that can be fatal for the procurement team, none have mentioned that health illiteracy-related experiences can affect the long-term effects of transplants [1,25]. Moreover, these trips frequently result in sleep deprivation due to their urgent nature, potentially impacting the health and performance of transplant surgeons. As highlighted by Englesbe and Merion, transplant surgeons often prioritize the urgency and significance of the organ procurement task, sometimes neglecting their safety [1].

Transplant surgeons interact with both cadaveric organ recipients and living organ transplantation patients and donors. As per regulations, surgeons refrain from engaging with the family members of cadaveric donors. Surgeons encounter various subcultures within these interactions, each with cultural norms, expectations, and attitudes toward health and healing. Through our interviews, we discovered that among the myriad of subcultural norms encountered in the realm of transplant surgery, particularly notable challenges arise with Jehovah’s Witnesses and Alevi-Bektashi patients. These groups exhibit distinct beliefs and practices that significantly impact their approach to medical treatment, including organ transplantation. Understanding and navigating these unique cultural perspectives is essential for transplant surgeons to provide adequate care and ensure optimal outcomes for patients from these backgrounds.

Jehovah’s Witnesses adhere to a prohibition on blood transfusions, prompting surgeons to devise a strategy of pre-storing patients’ blood before surgery. On the other hand, Alevi patients face challenges due to regulations that do not acknowledge ritual kinship. While numerous studies document surgeons’ preference for non-operative treatment for Jehovah’s Witnesses, this study marks the first exploration of the Alevi-Bektashi perspective on organ transplantation.

Despite transplant surgeons’ rigorous medical training, our findings highlight the limited emphasis on cultural competency within formal curriculums. Instead, surgeons predominantly acquire this essential skill through experiential learning and exposure to diverse patient populations. As articulated by Dr. Fatih, the complex interplay between culture and medical practice underscores the necessity for transplant surgeons to develop a nuanced understanding of cultural nuances to ensure optimal patient care. In conclusion, transplant surgeons navigate a landscape where their formal training equips them with theoretical expertise. However, it is in the daily encounters with patients and cultural challenges that their true excellence is tested and refined.

### 4.1. Limitations

Despite the comprehensive nature of this research, several limitations must be acknowledged. The sample size, with 21 kidney and liver transplant surgeons, may limit the generalizability of the findings. While there are 74 organ transplant centers in 9 RCCs, our data represent only cultural barriers faced by transplant surgeons working in 11 centers operating under the Ankara RCC. The issue of subcultural incompetency cannot be limited to Alevis and Jehovah’s Witnesses within Turkiye’s borders due to its diverse subcultural structure. Extending the research environment to the other eight RCCs would likely yield more comprehensive results that affect transplant surgery in general and transplant surgeons’ practices in particular. Cultural and religious sensitivities might have led to underreporting certain barriers due to social desirability bias. The focus on kidney and liver transplants potentially overlooks challenges unique to other specialties in transplant surgery, such as lung and heart. Addressing these limitations in future research will enhance the robustness and applicability of the findings, contributing to improved strategies for increasing organ donation rates and optimizing transplant outcomes.

### 4.2. Recommendations for Further Research

Based on the findings and limitations of this study, future research should expand its geographical scope to include all nine RCCs and their associated 74 organ transplant centers across Turkiye, ensuring a more comprehensive understanding of regional variations in cultural barriers. Increasing the sample size and diversity to include a broader range of transplant surgeons and coordinators would enhance the generalizability of the findings. Expanding the focus to other organ transplants beyond kidney and liver would provide a more holistic view of the challenges. Longitudinal studies are recommended to capture the evolving nature of cultural attitudes and practices. Additionally, studying public and media perceptions of organ donation and transplantation would offer insights into societal attitudes and help design effective communication strategies to promote organ donation.

### 4.3. Implications for Policy and Practice

The findings of this study on cultural barriers affecting transplant surgeons and patient care in Turkiye have significant implications for policy and practice. First, there is a clear need for targeted educational programs to raise awareness about organ donation and transplantation across diverse cultural and religious groups. These programs should be tailored to address specific misconceptions and fears prevalent within different subcultures. Additionally, expanding the scope of training for transplant surgeons and coordinators to include cultural competency is essential. Incorporating comprehensive cultural competency training into medical education and ongoing professional development will equip healthcare providers with the skills to navigate complex cultural dynamics and improve patient interactions. Policymakers should consider establishing and promoting support systems within medical institutions to address the emotional and psychological burdens transplant coordinators and surgeons face. Providing resources such as counseling services, peer support groups, and stress management programs can help mitigate burnout and improve job satisfaction, ultimately enhancing the overall effectiveness of transplant teams.

Moreover, enhancing public engagement through media campaigns that accurately represent the benefits and processes of organ donation and transplantation can help shift public perceptions and reduce stigma. Collaborating with influential community leaders and leveraging various media platforms can amplify positive messages and foster a more supportive environment for organ donation. By addressing these implications, policymakers and practitioners can develop and implement strategies that increase organ donation rates and enhance the overall quality of care provided to transplant patients and prospective donor families.

## 5. Conclusions

This study underscores the profound impact of cultural competency on the practice and success of transplant surgeons in Turkiye. Surgeons face unique challenges when dealing with deeply ingrained cultural beliefs, such as the religious trust placed in them by patients, the moral complexities of family dynamics during organ donation, and the health illiteracy that complicates organ procurement processes. The findings highlight the necessity for enhanced cultural competency training within medical education to better prepare surgeons for these challenges. Additionally, this study reveals that practical, hands-on experience with diverse patient populations is crucial for developing the cultural sensitivity needed to navigate these barriers effectively.

## Data Availability

The data represented in this paper are available from the corresponding author upon reasonable request.

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
