# Peer review of "Navigating Cultural Challenges in Transplant Surgery: Insights from Turkish Surgeons"

_healthcare, 2024, doi:10.3390/healthcare12131252_

Round 1

Reviewer 1 Report

Comments and Suggestions for Authors

Review manuscript: 3056308

I believe organ transplantation is often viewed as a great achievement in biomedicine, representing triumph over cultural barriers, however, transplant surgeons encounter various challenges, some of which are influenced by cultural factors. In the present study authors tried to explore the cultural challenges encountered by Turkish surgeons in the field of transplant surgery. To this end, the authors conducted semi-structured interviews with 21 Turkish surgeons to investigate the cultural factors influencing transplant surgery. The manuscript read well and appropriately discussed the cultural factors that influence the transplant surgeons. Despite addressing the challenges faced by transplant surgeons the present study requires a revision. Below are some of my specific comments:

  1. The study results have been limited to the 21 surgeons located in Ankara, Turkey’s capital. Given the relatively small sample size, I believe expanding the study to include surgeons from other regions of Turkey would enhance the quality and generalizability of the results.
  2. The abstract is missing details regarding the scope of the study. 
  3. The introduction section seems to be general and lacks pertinent background on the main objectives of the study. It would be beneficial to incorporate the significance of the study, explaining how the obtained results could improve patient care. Additionally, addressing the general challenges and barriers in organ donation and transplant surgery, and highlighting how studying cultural factors can help overcome these barriers, would provide valuable insights for the healthcare community.
  4. Elaborate on the methods sections to include the specific questions asked during the interview.
  5. Try to provide the recommendations in the study to overcome such potential cultural barriers.  

  1. Line 21: Missing Space;
  2. Methods and data section: The first para can be removed.
  3. Line 107: “X” University? Was it a typo?
  4. Sections 3-6, add them to the Results and Discussion sections.
  5. Lines 261-265 appears to be a repetition of the lines from 253-259. 
  6. Turkey has been presented in many formats. I would recommend changing all instances of ‘Turkey’, ‘Turkiye’, and ‘Türkiye’ to “Turkey” for consistency. 

Author Response

Review manuscript: 3056308

Navigating Cultural Challenges in Transplant Surgery: Insights from Turkish Surgeons

Dear Reviewer 1,

First, we would like to thank you for giving us another chance. We have been working on the manuscript day and night, and we hope not to disappoint you. We express our deep gratitude for your valuable time and insightful contributions, which have greatly improved the quality of our paper. We truly appreciate how thoroughly you read our paper and the effort you have put into enhancing it.

Second, we would like to present the revised matters individually through our responses: 

  1. The study results have been limited to the 21 surgeons in Ankara, Turkey’s capital. Given the relatively small sample size, expanding the study to include surgeons from other regions of Turkey would enhance the quality and generalizability of the results.

We have addressed your suggestion by restructuring the methods section. Our research primarily focuses on the cultural and folklore-related reasons for insufficient organ donations, particularly for kidneys and livers. Consequently, we conducted our fieldwork in Ankara, where 11 Organ Transplant Centers perform cadaveric and living organ transplants. Ankara serves as the central RCC and oversees 8 other RCCs throughout Türkiye. Additionally, the National Organ Allocation system is managed by the Ankara RCC. We also addressed your suggestion, pointing out the sampling size in the limitations sections.

You can review the revisions regarding the sampling size in the manuscript, highlighted in green:

“2.1.   Participants: We designed an ethnographic fieldwork study to understand the cultural dimensions of organ transplantation and the issue of insufficient organ donations. We engaged extensively with social actors whose lives revolve around organ transplantation and donation, including patients, caregivers of patients, donor families, organ procurement coordinators, and transplant surgeons. During our fieldwork from August 15, 2022, to October 15, 2023, we collaborated with 21 transplant surgeons specializing in kidney and liver transplantation. The National Organ Transplant Center in Ankara operates nine Regional Organ Coordination Centers (RCC) throughout Türkiye, including the Ankara RCC. There are ten organ transplant centers in Ankara where kidney and liver transplants are performed. “

“4.1.  Limitations: Despite the comprehensive nature of this research, several limitations must be acknowledged. The sample size, with 21 kidney and liver transplant surgeons, may limit the generalizability of the findings. While there are 74 organ transplant centers in 9 RCCs, our data represent only cultural barriers faced by transplant surgeons working in 11 centers operating under Ankara RCC. The issue of subcultural incompetency cannot be limited to Alevi and Jehovah's Witnesses within Türkiye's borders due to its diverse subcultural structure. Extending the research environment to the other eight RCCs would likely yield more comprehensive results that affect transplant surgery in general and transplant surgeons’ practices in particular.”

“4.2.   Recommendations for further research: Based on the findings and limitations of this study, future research should expand its geographical scope to include all nine RCCs and their associated 74 organ transplant centers across Türkiye, ensuring a more comprehensive understanding of regional variations in cultural barriers. Increasing the sample size and diversity to include a broader range of transplant surgeons and coordinators would enhance the generalizability of the findings.”

  1. The abstract is missing details regarding the scope of the study. 

We addressed your suggestion by rewriting the abstract in which we stated the scope of the study:

To achieve expertise, transplant surgeons in Türkiye undergo rigorous training, including medical school, residency, compulsory service, and extensive training in transplant surgery. Despite their high academic and clinical knowledge level, success in transplant surgery heavily depends on cultural competency. Through semi-structured interviews with 21 transplant surgeons specializing in kidney and liver transplants in Ankara, the study reveals how health illiteracy, culture, and folklore create significant barriers. Surgeons navigate these challenges while enduring harsh working conditions. The research highlights the critical role of cultural competency in transplant surgery, emphasizing the necessity for surgeons to understand and address the diverse cultural needs of their patients. Key findings indicate that surgeons must balance medical expertise with cultural sensitivity to deliver effective care. The study identifies four main cultural barriers: spiritual trust, family politics, health illiteracy, and subcultural incompetency. Effective transplant surgery requires a combination of theoretical proficiency and cultural awareness to meet patient's needs and improve surgical outcomes. “

  1. Elaborate on the methods sections to include the specific questions asked during the interview.

We addressed your suggestion by restructuring the method section.:

“2.2.     Data Collection: We facilitated semi-structured, in-depth, face-to-face interviews to explore whether surgeons encountered particularly challenging situations with their patients solely influenced by culture and folklore and how they handled these challenges to meet patient needs. Semi-structured interviews covered several themes: 1) How surgeons decide to pursue a career in transplant surgery, 2) How their profession affects their personal life, 3) How patients' literacy on health, particularly in the area of organ transplant, challenges their practice, 4) whether transplant surgeons encountered problems solely caused by social norms, folk beliefs, and religious views of patients, and 5) How they addressed these challenges. The informants' responses provided personal experience narratives, highlighting how cultural competency significantly impacts the success of transplant surgery. Additionally, we recorded observation notes at the end of each day spent in organ transplant centers. Our engagement with surgeons involved closely following their daily schedules, commencing at 6:30 in the morning and concluding when they departed from the hospital premises over a year. We accompanied them through clinical activities such as patient consultations, medical assessments, and interdisciplinary meetings and conducted in-depth interviews with healthcare providers and recipients.”

  1. Try to provide the recommendations in the study to overcome such potential cultural barriers.  

We addressed your suggestion by adding new sections:

“4.2. Recommendations for further research

Based on the findings and limitations of this study, future research should expand its geographical scope to include all nine RCCs and their associated 74 organ transplant centers across Türkiye, ensuring a more comprehensive understanding of regional var-iations in cultural barriers. Increasing the sample size and diversity to include a broader range of transplant surgeons and coordinators would enhance the generalizability of the findings. Expanding the focus to other organ transplants beyond kidney and liver would provide a more holistic view of the challenges. Longitudinal studies are recommended to capture the evolving nature of cultural attitudes and practices. Additionally, studying public and media perceptions of organ donation and transplantation would offer insights into societal attitudes and help design effective communication strategies to promote organ donation.

4.2.         Implications for policy and practice

The findings of this study on cultural barriers affecting transplant surgeons and patient care in Türkiye have significant implications for policy and practice. First, there is a clear need for targeted educational programs to raise awareness about organ donation and transplantation across diverse cultural and religious groups. These programs should be tailored to address specific misconceptions and fears prevalent within different subcul-tures. Additionally, expanding the scope of training for transplant surgeons and coor-dinators to include cultural competency is essential. Incorporating comprehensive cultural competency training into medical education and ongoing professional development will equip healthcare providers with the skills to navigate complex cultural dynamics and improve patient interactions. Policymakers should consider establishing and promoting support systems within medical institutions to address the emotional and psychological burdens transplant coordinators and surgeons face. Providing resources such as coun-seling services, peer support groups, and stress management programs can help mitigate burnout and improve job satisfaction, ultimately enhancing the overall effectiveness of transplant teams.

Moreover, enhancing public engagement through media campaigns that accurately represent the benefits and processes of organ donation and transplantation can help shift public perceptions and reduce stigma. Collaborating with influential community leaders and leveraging various media platforms can amplify positive messages and foster a more supportive environment for organ donation. By addressing these implications, policy-makers and practitioners can develop and implement strategies that increase organ donation rates and enhance the overall quality of care provided to transplant patients and prospective donor families."

  1. Line 21: Missing Space

We have eliminated all missing spaces.

  1. Methods and data section: The first para can be removed.

We removed the first paragrapgh.

  1. Line 107: “X” University? Was it a typo?

We had to cover the name of university to not harm the peer review process. You will able to see the name of institution where we obtain our ethical approval in ethic consideration section:

2.4.         Ethical Considerations: Due to the highly personal nature of the interview contexts, strict confidentiality measures were implemented for our informants. The actual identities of the surgeons, as documented on consent forms and audio recordings, are known solely to the research team. To safeguard anonymity, all real names of informants have been replaced with common Turkish names, ensuring confidentiality between the research team and participants. By providing anonymity, our informants felt free to express their thoughts, feelings, and personal experiences, which may have influenced their performance. Furthermore, in line with our ethical obligations, we took measures to safeguard their social and economic well-being during and after the fieldwork. Studies have demonstrated that ensuring anonymity for informants in qualitative studies not only fosters trust between researchers and participants but also encourages informants to share their experiences without fear of repercussions related to the research topic (11,14). Since most participating surgeons voluntarily engaged in our research within state-owned hospitals and held academic positions, our study posed no threat to their well-being. Before embarking on fieldwork, ethics committee approval was obtained from Bartın University Social and Humanities Research Ethics Board, 2022 SBB-0055, on February 22, 2022.

  1. Sections 3-6, add them to the Results and Discussion sections.

Following your suggestion we have added results and discussion sections:

  1. Results: The findings of this study highlight several critical cultural barriers. The first central theme is spiritual trust, where many patients express profound trust in their surgeons, often invoking religious beliefs by stating, "I entrust my life first to Allah and then to you." This expression adds emotional weight to the surgeons' responsibilities and increases their stress during transplant operations. Most respondents (n=18) navigate this emotional burden by placing their faith in Allah for the outcome while performing to the best of their abilities, with only a minority (n=2) engaging in discussions about potential outcomes with patients during the consent process to manage this stress. Family politics also play a crucial role, with most respondents (n=20) expressing frustration over fully informed and consented prospective donors expecting surgeons to liberate them from donation without disclosing the truth to the prospective recipient. This dynamic is particularly challenging when female family members are expected to donate, leading surgeons (n=13) to take extra care to ensure donors are fully informed and not coerced. Health illiteracy among patients and the general public presents significant challenges, as all interviewees (n=21) noted patients' tendencies to trust unreliable sources over medically accurate knowledge. This leads to unrealistic expectations and misunderstandings. Health illiteracy also complicates organ procurement trips, with several informants (n=11) highlighting issues during commercial airplane journeys. Lastly, subcultural incompetency is evident as surgeons encounter various subcultural norms impacting their practice. Jehovah’s Witnesses refuse blood transfusions due to religious beliefs, compelling surgeons (n=6) to devise strategies like pre-storing patients' blood.
    In contrast, Alevi-Bektashi patients face challenges related to ritual kinship, which is often unrecognized by legal regulations. The majority of informants (n=11) indicated that Alevi patients felt offended when doctors questioned their ritual brotherhood. These subcultural nuances necessitate surgeons to deeply understand their patients' cultural backgrounds to provide adequate care. The study underscores that cultural competency is crucial for transplant surgery success, often gained through practical experience rather than formal training, highlighting the need for ongoing cultural education in medical practice.
  2. Discussion: Given the cultural diversity in Türkiye, physicians regularly interact with patients from various ethnic backgrounds and cultural heritages. The effectiveness of medical care is not solely dependent on the theoretical proficiency of practitioners but also on their ability to address the cultural needs of patients. Culture shapes patients' expectations and influences their attitudes and behaviors towards treatment. Western medical training often assumes that all patients will be highly educated and compliant with prescribed medicines. However, doctors encounter many diverse culture-nurtured scenarios in their daily practice. Transplant surgery is a distinctive medical specialty wherein extensively trained surgeons perform the pivotal task of replacing malfunctioning organs with healthy counterparts. This transformative intervention bestows renewed vitality upon patients and is a beacon of hope, easing their burdensome afflictions. While extensive research has delved into the professional challenges faced by transplant surgeons, only a limited number of studies have explored whether these professionals encounter difficulties arising from the cultural needs of their patients (4,6). To address this gap, we augmented our semi-structured, face-to-face interviews with transplant surgeons by incorporating additional inquiries to investigate the potential correlation between the cultural competency of surgeons and the success of transplant surgery. Our results came up with four teams: Spiritual trust, family politics, health illiteracy, and subcultural incompetency.

Central to our study is the profound trust that patients place in transplant surgeons, often invoking religious beliefs and entrusting their lives to both Allah and the physician. This expression of trust not only amplifies the emotional responsibility felt by surgeons but also shapes their coping mechanisms and approach to patient care. Dr. Ahmet's perspective exemplifies the delicate balance between professional competency and reliance on divine intervention, highlighting the existential uncertainties inherent in surgical practice.

While many have highlighted the moral obligation placed on family members in the context of organ transplantation to donate spare organs, there has been a notable absence of mention regarding the involvement of transplant surgeons in family politics to alleviate this emotional burden from morally obligated prospective donors (16,17,18). Turkish transplant surgeons are dragged into family politics by reluctant living organ donors who are afraid of confrontation or being marginalized by family members by labeling them with selfishness. As Elliot pointed out, a physician's role entails assessing the reasonableness or moral justifiability of a patient's decision and determining if the physician is morally justified in aiding the patient in achieving it (42). In such instances, the participation of surgeons in family politics is geared toward safeguarding hesitant living organ donors. Moreover, as moral agents, surgeons reserve the right to decline a donor if the associated risks are deemed excessively high, particularly in emotionally charged donation scenarios (17,18). Transplant surgery's complexity is often misunderstood by those outside the medical field, with layman's knowledge largely shaped by urban narratives and fictional movies (43,44,45). Dr. Ali’s accounts illustrate how medical professionals outside the transplant surgery may have difficulty understanding the details of transplant surgery. The impacts of health illiteracy are mostly traceable through the personal experience of transplant surgeons on organ procurement trips. Even though studies have underlined the risks of these trips that might be fatal for the procurement team, none have mentioned that health illiteracy-related experiences could affect the long-term effects of transplants (1,25). Moreover, these trips frequently result in sleep deprivation due to their urgent nature, potentially impacting the health and performance of transplant surgeons. As highlighted by Englesbe and Merion, transplant surgeons often prioritize the urgency and significance of the organ procurement task, sometimes neglecting their safety (1).

Transplant surgeons interact with both cadaveric organ recipients and living organ transplantation patients and donors. As per regulations, surgeons refrain from engaging with the family members of cadaveric donors. Surgeons encounter various subcultures within these interactions, each with cultural norms, expectations, and attitudes toward health and healing. Through our interviews, we discovered that among the myriad of subcultural norms encountered in the realm of transplant surgery, particularly notable challenges arise with Jehovah’s Witnesses and Alevi-Bektashi patients. These groups exhibit distinct beliefs and practices that significantly impact their approach to medical treatment, including organ transplantation. Understanding and navigating these unique cultural perspectives is essential for transplant surgeons to provide adequate care and ensure optimal outcomes for patients from these backgrounds.

Jehovah’s Witnesses adhere to a prohibition on blood transfusions, prompting surgeons to devise a strategy of pre-storing patients' blood before surgery. On the other hand, Alevi patients face challenges due to regulations that do not acknowledge ritual kinship. While numerous studies document surgeons' preference for non-operative treatment for Jehovah’s Witnesses, this study marks the first exploration of the Alevi-Bektashi perspective on organ transplantation. Despite transplant surgeons' rigorous medical training, our findings highlight the limited emphasis on cultural competency within formal curriculums. Instead, surgeons predominantly acquire this essential skill through experiential learning and exposure to diverse patient populations. As articulated by Dr. Fatih, the complex interplay between culture and medical practice underscores the necessity for transplant surgeons to develop a nuanced understanding of cultural nuances to ensure optimal patient care. In conclusion, transplant surgeons navigate a landscape where their formal training equips them with theoretical expertise. However, it is in the daily encounters with patients and cultural challenges that their true excellence is tested and refined.

  1. Lines 261-265 appear to be a repetition of the lines from 253-259. 

Following your suggestion, we have removed the repetition:

Housewives, in particular, are expected to step forward as donors since they are not employed, and other family members can assist them during their recovery period. When potential donors were female, transplant surgeons exercised extreme caution to ensure that they were not being coerced into donation. Most of our informants (n=13) indicated that when patients brought their sisters-in-law, daughters-in-law, sisters, wives, or mothers as potential donors, they ensured that the donor was fully informed and willing to undergo the donation process. Dr. Oskay said:

When the potential donor is female, we must proceed cautiously. I pay close attention to her demeanor during family meetings; her jests, facial expressions, and subtle cues often reveal her reluctance. In such cases, I provide both the recipient and donor with my business card, encouraging the donor to reach out and share her thoughts with me. More often than not, they contact me to express concerns about feeling pressured or fearful of offending the patient. In response, we work together to navigate a solution that respects her wishes and saves her from the donation process.

It is well-known that in cost-free living donations, two-thirds of all organs are donated by women. Studies related this to women's sense of responsibility and self-sacrifice tendency (21,22). Dr. Ulvi’s experiences with female donors suggest that parenting female members of the family who are housewives are first to be expected as organ donors: “Male patients behave as if their wives and elder sisters have to sacrifice their kidneys or livers for their brothers and husbands.

  1. Turkey has been presented in many formats. I would recommend changing all instances of ‘Turkey’, ‘Turkiye’, and ‘Türkiye’ to “Turkey” for consistency.

We have addressed your suggestion, reassuring the consistency: Türkiye

We hope our response and revised paper version will meet your expectations. We sincerely appreciate your time and feedback. Best regards,

Corresponding Author of the Manuscript Healthcare_3056308

Reviewer 2 Report

Comments and Suggestions for Authors

The article fits well within the thematic scope of the journal and, as such, may contribute to the literature. However, several changes need to be made and missing information needs to be added:

Line 61: The authors wrote: "I watched an elderly lady..." Shouldn't the sentence start with "We"? If the first sentence is part of a quote, quotation marks should be inserted.

Lines 61-72: It is not clear what the purpose of the description in this section of the article is (lines 61-72). In a section titled Methods and Data, one would expect a description of specific research methods.

There is no information on how many participants took part in the study. Additionally, it is not clear how the respondents were selected (randomly or according to some criteria)? (lines 109-119)

The article does not contain a Limitations section. It must be emphasized that there is no research without limitations. The research presented by the authors has its limitations. First, it is not known how many respondents there were. Second, the research is limited to 11 hospitals in Ankara. The research may not be representative of the entire country of Turkey. Third, further research is needed - this should be emphasized in the article - to compare the trends indicated in the article with other centers/hospitals in the country.

Author Response

Review manuscript: 3056308

Navigating Cultural Challenges in Transplant Surgery: Insights from Turkish Surgeons

Dear Reviewer 2,

First, we would like to thank you for giving us another chance. We have been working on the manuscript day and night, and we hope not to disappoint you. We express our deep gratitude for your valuable time and insightful contributions, which have greatly improved the quality of our paper. We truly appreciate how thoroughly you read our paper and the effort you have put into enhancing it. We are especially grateful for your detailed feedback and the constructive suggestions you provided. Your expertise and thoughtful comments have highlighted areas for improvement and guided us toward refining our arguments and strengthening our analysis. Your meticulous care in reviewing our manuscript reflects your dedication to academic excellence, and we are honored to have benefited from your knowledge and experience. Second, we would like to present the revised matters individually through our response:

The article fits well within the thematic scope of the journal and, as such, may contribute to the literature. However, several changes need to be made and missing information needs to be added:

Line 61: The authors wrote: "I watched an elderly lady..." Shouldn't the sentence start with "We"? If the first sentence is part of a quote, quotation marks should be inserted.

Following the suggestion of reviewer 1, we removed the paragraph.

Lines 61-72: It is not clear what the purpose of the description in this section of the article is (lines 61-72). In a section titled Methods and Data, one would expect a description of specific research methods. There is no information on how many participants took part in the study. Additionally, it is not clear how the respondents were selected (randomly or according to some criteria)? (lines 109-119)

We addressed the revision by restructuring the method section, highlighted in red in the manuscript:

  1. Method

2.1.         Participants

We designed an ethnographic fieldwork study to understand the cultural dimensions of organ transplantation and the issue of insufficient organ donations. We engaged extensively with social actors whose lives revolve around organ transplantation and donation, including patients, caregivers of patients, donor families, organ procurement coordinators, and transplant surgeons. During our fieldwork from August 15, 2022, to October 15, 2023, we collaborated with 21 transplant surgeons specializing in kidney and liver transplantation. The National Organ Transplant Center in Ankara operates nine Regional Organ Coordination Centers (RCC) throughout Türkiye, including the Ankara RCC. There are ten organ transplant centers in Ankara where kidney and liver transplants are performed.

Most of our participants (n=18) were affiliated with state-owned organ transplant centers, while the remaining were associated with organ transplant centers linked to private university hospitals. Among our participants, a significant portion (n=16) held academic positions, primarily as professors and associates, with a few others (n=3) progressing in their academic careers. The age range of our participants varied, with the youngest being 37 years old and the eldest being 70 years old. Most respondents were male (n=18), with a smaller proportion being female (n=3). Regarding marital status, most participants (n=19) were married, while two were divorced. All informants were parents.

2.2.         Data Collection

We facilitated semi-structured, in-depth, face-to-face interviews to explore whether surgeons encountered particularly challenging situations with their patients solely influenced by culture and folklore and how they handled these challenges to meet patient needs. Semi-structured interviews covered several themes: 1) How surgeons decide to pursue a career in transplant surgery, 2) How their profession affects their personal life, 3) How patients' literacy on health, particularly in the area of organ transplant, challenges their practice, 4) whether transplant surgeons encountered problems solely caused by social norms, folk beliefs, and religious views of patients, and 5) How they addressed these challenges. The informants' responses provided personal experience narratives, highlighting how cultural competency significantly impacts the success of transplant surgery. Additionally, we recorded observation notes at the end of each day spent in organ transplant centers. Our engagement with surgeons involved closely following their daily schedules, commencing at 6:30 in the morning and concluding when they departed from the hospital premises over a year. We accompanied them through clinical activities such as patient consultations, medical assessments, and interdisciplinary meetings and conducted in-depth interviews with healthcare providers and recipients.

2.3.         Data Analysis

All recorded interviews were decoded via Transcriptor and listened to by the research team to make required corrections on transcripts. Throughout translations of transcripted records, we meticulously reflected our informants' cultural and emotional reflections. The responses of informants and observation notes regarding interview questions matched. After reading transcripts from several interviews, the authors developed a coding scheme to capture the essence of the testimonials using MAXQDA 2024. We facilitated thematic analysis to identify and report the patterns that emerged in our data. The theme of cultural barriers emerged with four sub-themes: spiritual trust, familial politics, health illiteracy, and subcultural competency.

2.4.         Ethical Considerations

Due to the highly personal nature of the interview contexts, strict confidentiality measures were implemented for our informants. The actual identities of the surgeons, as documented on consent forms and audio recordings, are known solely to the research team. To safeguard anonymity, all real names of informants have been replaced with common Turkish names, ensuring confidentiality between the research team and participants. By providing anonymity, our informants felt free to express their thoughts, feelings, and personal experiences, which may have influenced their performance.

Furthermore, in line with our ethical obligations, we took measures to safeguard their social and economic well-being during and after the fieldwork. Studies have demonstrated that ensuring anonymity for informants in qualitative studies not only fosters trust between researchers and participants but also encourages informants to share their experiences without fear of repercussions related to the research topic (11,14). Since most participating surgeons voluntarily engaged in our research within state-owned hospitals and held academic positions, our study posed no threat to their well-being. Before embarking on fieldwork, ethics committee approval was obtained from Bartın University Social and Humanities Research Ethics Board, 2022 SBB-0055, on February 22, 2022

The article does not contain a Limitations section. It must be emphasized that there is no research without limitations. The research presented by the authors has its limitations. First, it is not known how many respondents there were. Second, the research is limited to 11 hospitals in Ankara. The research may not be representative of the entire country of Turkey. Third, further research is needed - this should be emphasized in the article - to compare the trends indicated in the article with other centers/hospitals in the country.

We agree that the paper must include a limitations section. Therefore, we have added several sections addressing this.

4.1.         Limitations

Despite the comprehensive nature of this research, several limitations must be acknowledged. The sample size, with 21 kidney and liver transplant surgeons, may limit the generalizability of the findings. While there are 74 organ transplant centers in 9 RCCs, our data represent only cultural barriers faced by transplant surgeons working in 11 centers operating under Ankara RCC. The issue of subcultural incompetency cannot be limited to Alevi and Jehovah's Witnesses within Türkiye's borders due to its diverse subcultural structure. Extending the research environment to the other eight RCCs would likely yield more comprehensive results that affect transplant surgery in general and transplant surgeons’ practices in particular. Cultural and religious sensitivities might have led to underreporting certain barriers due to social desirability bias. The focus on kidney and liver transplants potentially overlooks challenges unique to other specialties in transplant surgery, such as lung and heart. Addressing these limitations in future research will enhance the robustness and applicability of the findings, contributing to improved strategies for increasing organ donation rates and optimizing transplant outcomes.

4.2. Recommendations for further research

Based on the findings and limitations of this study, future research should expand its geographical scope to include all nine RCCs and their associated 74 organ transplant centers across Türkiye, ensuring a more comprehensive understanding of regional variations in cultural barriers. Increasing the sample size and diversity to include a broader range of transplant surgeons and coordinators would enhance the generalizability of the findings. Expanding the focus to other organ transplants beyond kidney and liver would provide a more holistic view of the challenges. Longitudinal studies are recommended to capture the evolving nature of cultural attitudes and practices. Additionally, studying public and media perceptions of organ donation and transplantation would offer insights into societal attitudes and help design effective communication strategies to promote organ donation.

4.2.         Implications for policy and practice

The findings of this study on cultural barriers affecting transplant surgeons and patient care in Türkiye have significant implications for policy and practice. First, there is a clear need for targeted educational programs to raise awareness about organ donation and transplantation across diverse cultural and religious groups. These programs should be tailored to address specific misconceptions and fears prevalent within different subcul-tures. Additionally, expanding the scope of training for transplant surgeons and coor-dinators to include cultural competency is essential. Incorporating comprehensive cultural competency training into medical education and ongoing professional development will equip healthcare providers with the skills to navigate complex cultural dynamics and improve patient interactions. Policymakers should consider establishing and promoting support systems within medical institutions to address the emotional and psychological burdens transplant coordinators and surgeons face. Providing resources such as coun-seling services, peer support groups, and stress management programs can help mitigate burnout and improve job satisfaction, ultimately enhancing the overall effectiveness of transplant teams.

Moreover, enhancing public engagement through media campaigns that accurately represent the benefits and processes of organ donation and transplantation can help shift public perceptions and reduce stigma. Collaborating with influential community leaders and leveraging various media platforms can amplify positive messages and foster a more supportive environment for organ donation. By addressing these implications, policy-makers and practitioners can develop and implement strategies that increase organ donation rates and enhance the overall quality of care provided to transplant patients and prospective donor families.

We are grateful for the time you took to read this work, for sharing your wisdom and expertise to improve our manuscript, and for the humility with which you did so. Considering how rare it is to find experts who can provide constructive criticism without causing distress, we particularly want to thank you for your scientific and humane language. Sincerely yours

Corresponding Author of the Manuscript Healthcare_3056308

Round 2

Reviewer 1 Report

Comments and Suggestions for Authors

Thank you for your responses.

Reviewer 2 Report

Comments and Suggestions for Authors

I have no more comments.